# Optimistic Natural Policy Gradient: a Simple Efficient Policy Optimization Framework for Online RL

**Qinghua Liu**[*]
Princeton University
qinghual@princeton.edu

**Gellért Weisz**
Google DeepMind
gellert@deepmind.com

**András György**
Google DeepMind
agyorgy@deepmind.com

**Chi Jin**
Princeton University
chij@princeton.edu

**Csaba Szepesvári**
Google DeepMind and University of Alberta
szepesva@ualberta.ca

## Abstract

While policy optimization algorithms have played an important role in recent empirical success of Reinforcement Learning (RL), the existing theoretical understanding of policy optimization remains rather limited—they are either restricted to tabular MDPs or suffer from highly suboptimal sample complexity, especially in online RL where exploration is necessary. This paper proposes a simple efficient policy optimization framework—OPTIMISTIC NPG for online RL. OPTIMISTIC NPG can be viewed as a simple combination of the classic natural policy gradient (NPG) algorithm [Kakade, 2001] and an optimistic policy evaluation subroutine to encourage exploration. For $d$-dimensional linear MDPs, OPTIMISTIC NPG is computationally efficient, and learns an $\epsilon$-optimal policy within $\tilde{\mathcal{O}}(d^2/\epsilon^3)$ samples, which is the first computationally efficient algorithm whose sample complexity has the optimal dimension dependence $\tilde{\Theta}(d^2)$. It also improves over state-of-the-art results of policy optimization algorithms [Zanette et al., 2021] by a factor of $d$. For general function approximation that subsumes linear MDPs, OPTIMISTIC NPG, to our best knowledge, is also the first policy optimization algorithm that achieves the polynomial sample complexity for learning near-optimal policies.

## 1 Introduction

Policy optimization algorithms [Schulman et al., 2017, 2015] with neural network function approximation have played an important role in recent empirical success of reinforcement learning (RL), such as robotics [Finn et al., 2016], games [Berner et al., 2019] and large language models [OpenAI, 2022]. Motivated by the empirical success, the theory community made a large effort to design provably efficient policy optimization algorithms that work in the presence of linear function approximation [Agarwal et al., 2021, Bhandari and Russo, 2019, Liu et al., 2019, Neu et al., 2017, Abbasi-Yadkori et al., 2019, Agarwal et al., 2020, Zanette et al., 2021, Shani et al., 2020, Cai et al., 2020]. Early works focused on proving that policy optimization algorithms are capable to learn near-optimal policies using a polynomial number of samples under certain reachability (coverage) assumptions. [e.g., Agarwal et al., 2021, Bhandari and Russo, 2019, Liu et al., 2019, Neu et al., 2017, Abbasi-Yadkori et al., 2019]. While this was good for laying down the foundations for future work, the reachability assumptions basically imply that the state space is already well-explored or rather easy to explore, which avoids the challenge of performing strategic exploration — one of the central problems in both empirical and theoretical RL.

---

[*]This work was done when QL interned at DeepMind.

37th Conference on Neural Information Processing Systems (NeurIPS 2023).

Table 1: A comparison of sample-complexity results for linear MDPs.

| Algorithms | Sample Complexity | Computationally Efficient | Policy Optimization |
|---|---|---|---|
| Zanette et al. [2020] Jin et al. [2021] | $\tilde{\Theta}(d^2/\epsilon^2)$ | × | × |
| He et al. [2022] Agarwal et al. [2022] Wagenmaker et al. [2022] | $\tilde{\mathcal{O}}(d^2/\epsilon^2 + d^{\geq 4}/\epsilon)$ | ✓ | × |
| Agarwal et al. [2020] | $\tilde{\mathcal{O}}(\text{poly}(d)/\epsilon^{11})$ | ✓ | ✓ |
| Zanette et al. [2021] | $\tilde{\mathcal{O}}(d^3/\epsilon^3)$ | ✓ | ✓ |
| Optimistic NPG (**this work**) | $\tilde{\mathcal{O}}(d^2/\epsilon^3)$ | ✓ | ✓ |

To address this limitation, later works [Agarwal et al., 2020, Zanette et al., 2021] proposed policy optimization algorithms that enjoy polynomial sample-complexity guarantee without making any reachability assumption, but at the cost of either complicating the algorithm design (and analysis) with various tricks or getting highly suboptimal sample-complexity guarantees. For example, the PC-PG algorithm [Agarwal et al., 2020], which, to our knowledge was the first policy optimization algorithm for learning linear MDPs without reachability assumptions, requires $\tilde{\mathcal{O}}(\text{poly}(d)/\epsilon^{11})$ samples to learn an $\epsilon$-optimal policy for $d$-dimensional linear MDPs. That $\tilde{\mathcal{O}}(1/\epsilon^{11})$ samples were necessary for this task is highly unlikely. Indeed, Zanette et al. [2021] greatly improved this sample complexity to $\tilde{\mathcal{O}}(d^3/\epsilon^3)$ at the cost of considerably complicating the algorithm design and the analysis. Moreover, we are not aware of efficient guarantees of policy optimization for generic function approximation, which is rich enough to subsume linear MDPs. This motivates us to ask the following question:

*Can we design a **simple, general** policy optimization algorithm with **sharp** sample-complexity guarantees in the **exploration** setting[2]?*

In particular, we aim to achieve sharper sample complexity than Zanette et al. [2021] in the linear MDP setting, and achieve low-degree polynomial sample complexity in the general function approximation setting. This paper answers the highlighted question affirmatively by making the following contributions:

- **Sharper rate.** We propose a computationally efficient policy optimization algorithm—OPTIMISTIC NPG with sample complexity

$$\tilde{\mathcal{O}}(d^2/\epsilon^3)$$

for learning an $\epsilon$-optimal policy in an online fashion while interacting with a $d$-dimensional linear MDP. This result improves over the best previous one [Zanette et al., 2021] in policy optimization by a factor of $d$. Moreover, to our knowledge, this is the first computationally efficient algorithm to achieve the optimal quadratic dimension dependence. Before moving on, we remark that previous FQI-style algorithms [Zanette et al., 2020, Jin et al., 2021] achieve the optimal sample complexity $\tilde{\Theta}(d^2/\epsilon^2)$ [Zanette et al., 2020] but they are computationally *inefficient* due to the mechanism of global optimism in their algorithm design. Several very recent works [He et al., 2022, Agarwal et al., 2022, Wagenmaker et al., 2022] achieve the sample complexity $\tilde{\mathcal{O}}(d^2/\epsilon^2 + d^{\geq 4}/\epsilon)$ using computationally efficient algorithms. Compared to our work, the aforementioned rate has worse dependence on $d$ and better dependence on $\epsilon$. Nonetheless, our result is better in the practically important regime where the feature dimension $d$ is typically very large and the target accuracy $\epsilon$ is not too small. To summarize, the sample complexity of OPTIMISTIC NPG strictly improves over the best existing policy optimization algorithms and is not dominated by that of any existing computationally efficient algorithm.

---

[2]By "exploration setting", we mean a setup where there is no simulator nor reachability assumption and a learner can only influence the evolution of environmental states by taking actions.

To our best knowledge, this paper also achieves the first polynomial sample-complexity guarantee for policy optimization under general function approximation.

- **Simple on-policy algorithm.** At a high level, OPTIMISTIC NPG is almost an *on-policy* version of natural policy gradient (NPG) [Kakade, 2001] with Boltzmann policies that use linear function approximation for optimistic action-value estimates. By "on-policy", we mean the transition-reward data used to improve the current policy are obtained exactly by executing the current policy. However, the algorithm has a tuneable parameter (denoted by $m$ in Algorithm 1) that allows it to reuse data sampled from earlier policies. By using such data, the algorithm becomes *off-policy*. The optimal choice of the tuneable parameter is dictated by theoretical arguments. Interestingly, our analysis shows that even the purely on-policy version of the algorithm (i.e., set $m = 1$ in Algorithm 1) enjoys a well-controlled sample complexity of $\tilde{\mathcal{O}}(d^2/\epsilon^4)$. To the best of our knowledge, this is the first time an on-policy method is shown to have polynomial sample complexity in the exploration setting, given that all previous policy optimization or Q-learning style or model-based algorithms [e.g., Jin et al., 2020, Zanette et al., 2020, 2021, Agarwal et al., 2020, Cai et al., 2020, Shani et al., 2020, etc] are off-policy.

- **New proof techniques.** To achieve the improved rate, we introduce several new ideas in the proof, including but not limited to (a) exploiting the softmax parameterization of the policies to reuse data and improve sample efficiency (Lemma 3), (b) controlling *on-policy* uncertainty (Lemma 4) instead of cumulative uncertainty as in previous off-policy works, and (c) using a bonus term that is smaller than those in previous works by a factor of $\sqrt{d}$ (see Lemma 2 and the discussions that follow).

## 1.1 Related works

Since this paper studies policy optimization algorithms in the setting of linear MDPs, below we restrict our focus to previous theoretical works on either policy optimization or linear MDPs.

**Policy optimization.** This work is inspired by and builds upon two recent works [Shani et al., 2020, Cai et al., 2020] that combine NPG with bonus-based optimism to handle exploration. In terms of algorithmic design, this paper (OPTIMISTIC NPG) utilizes on-policy fresh data for value estimation while Shani et al. [2020], Cai et al. [2020] are off-policy and reuse *all* the historical data. In terms of theoretical guarantees, Shani et al. [2020] and Cai et al. [2020] only study tabular MDPs and linear mixture MDPs [Zhou et al., 2021] respectively, while this paper considers the more challenging setting of linear MDPs [Jin et al., 2020] (in our view, linear MDPs are more challenging as there the number of model parameters scale with the number of states). We remark that due to some subtle technical challenge, so far it still remains unknown whether it is possible to generalize the analysis of Cai et al. [2020] to handle linear MDPs. Agarwal et al. [2020], Zanette et al. [2021] derive the first line of policy optimization results for RL in linear MDPs without any reachability- (or coverage-) style assumption. Compared to Agarwal et al. [2020], Zanette et al. [2021], our work considers the same setting but designs a simpler algorithm with cleaner analysis and sharper sample-complexity guarantee. Nevertheless, we remark that in certain model-misspecification settings, the algorithms of Agarwal et al. [2020], Zanette et al. [2021] can potentially achieve stronger guarantees. Since we focus on the well-specified setting (that is, the environment is perfectly modeled by a linear MDP), we refer interested readers to Agarwal et al. [2020] for more details. Finally, there have been a long line of works [e.g., Agarwal et al., 2021, Bhandari and Russo, 2019, Liu et al., 2019, Neu et al., 2017, Abbasi-Yadkori et al., 2019, etc] that study policy optimization under reachability (or coverage) assumptions, which eliminates the need for performing strategic exploration. Throughout this paper we do not make any such assumption and directly tackle the exploration challenge.

**Linear MDPs.** For linear MDPs, Jin et al. [2020] proposed a computationally efficient algorithm (LSVI-UCB) with $\tilde{\mathcal{O}}(d^3/\epsilon^2)$ sample complexity. Later on, Zanette et al. [2020] utilized the idea of global optimism to obtain the optimal sample complexity $\tilde{\Theta}(d^2/\epsilon^2)$ at the cost of sacrificing computational efficiency. Recently, He et al. [2022], Agarwal et al. [2022], Wagenmaker et al. [2022] designed new computationally efficient algorithms that can achieve $\tilde{\mathcal{O}}(d^2/\epsilon^2 + d^{\geq 4}/\epsilon)$ sample complexity. Compared to the above works, the sample complexity of OPTIMISTIC NPG is not strictly worse than that of any known computationally efficient algorithm. In fact, it is the only computationally efficient method to achieve the optimal dimension dependence. Nonetheless, for

learning a near-optimal policy with a vanishing suboptimality $\epsilon$, the current sample complexity of OPTIMISTIC NPG is loose by a factor of $\epsilon^{-1}$.

## 2 Preliminaries

**MDP.** We consider the model of episodic Markov Decision Processes (MDPs). Formally, an MDP is defined by a tuple $(\mathcal{S}, \mathcal{A}, \mathbb{P}, R, H)$ where $\mathcal{S}$ denotes the set of states, $\mathcal{A}$ denotes the set of actions, both of which are assumed to be finite, $\mathbb{P} = \{\mathbb{P}_h\}_{h\in[H]}$ denotes the set of transition probability functions so that $\mathbb{P}_h(s' \mid s, a)$ is equal to the probability of transitioning to state $s'$ given that action $a$ is taken at state $s$ and step $h$, $R = \{R_h\}_{h\in[H]}$ denotes the collection of expected reward functions so that $R_h(s, a)$ is equal to the expected reward to be received if action $a$ is taken at state $s$ and step $h$, and $H$ denotes the length of each episode. An agent interacts an MDP in the form of episodes. Formally, we assume without loss of generality that each episode always starts from a *fixed* initial state $s_1$. At the $h^{\text{th}}$ step of this episode, the agent first observes the current state $s_h$, then takes action $a_h$ and receives reward $r_h(s_h, a_h)$ satisfying

$$\mathbb{E}[r_h(s_h, a_h) \mid s_h, a_h] = R_h(s_h, a_h).$$

After that, the environment transitions to

$$s_{h+1} \sim \mathbb{P}_h(\cdot \mid s_h, a_h).$$

The current episode terminates immediately once $r_H$ is received. Throughout this paper, we assume $\mathbb{P}$ and $R$ are unknown to the learner.

**Linear MDP.** A $d$-dimensional linear MDP [Jin et al., 2020] is defined by two sets of feature mappings, $\{\phi_h\}_{h\in[H]} \subseteq (\mathcal{S} \times \mathcal{A} \to \mathbb{R}^d)$ and $\{\psi_h\}_{h\in[H]} \subseteq (\mathcal{S} \to \mathbb{R}^d)$, and a set of vectors $\{w_h^\star\}_{h\in[H]} \subseteq \mathbb{R}^d$ so that the transition probability functions can be represented as bilinear functions of feature mappings $\{\phi_h\}_{h\in[H]}$ and $\{\psi_h\}_{h\in[H]}$, and the reward functions can be represented as linear functions of $\{\phi_h\}_{h\in[H]}$. Formally, we have that for all $(s, a, s') \in \mathcal{S} \times \mathcal{A} \times \mathcal{S}$:

$$\mathbb{P}_h(s' \mid s, a) = \langle \phi_h(s, a), \psi_h(s') \rangle,$$
$$R_h(s, a) = \langle \phi_h(s, a), w_h^\star \rangle.$$

For the purpose the regularity, linear MDPs also require that

$$\max_{h,s,a} \|\phi_h(s, a)\|_2 \le 1, \quad \max_h \|w_h^\star\|_2 \le \sqrt{d},$$

and for any function $V_{h+1} : \mathcal{S} \to [0, 1]$,

$$\left\| \int_{s\in\mathcal{S}} V_{h+1}(s)\psi_{h+1}(s)ds \right\|_2 \le \sqrt{d}.$$

Throughout this paper, we assume only $\{\phi_h\}_{h\in[H]}$ is available to the learner while $\{\psi_h\}_{h\in[H]}$ and $\{w_h^\star\}_{h\in[H]}$ are not.

**Policy and value.** A (Markov) policy is a set of conditional probability functions $\pi = \{\pi_h\}_{h\in[H]}$ so that $\pi_h(\cdot \mid s) \in \Delta_\mathcal{A}$ gives the distribution over the action set conditioned on the current state $s$ at step $h$. We define the V-value functions of policy $\pi$ by $\{V_h^\pi\}_{h\in[H]} \subseteq (\mathcal{S} \to \mathbb{R})$ so that $V_h^\pi(s)$ is equal to the expected cumulative reward an agent will receive if she follows policy $\pi$ starting from state $s$ and step $h$. Formally,

$$V_h^\pi(s) = \mathbb{E}\left[ \sum_{h'=h}^H r_{h'}(s_{h'}, a_{h'}) \mid s_h = s, a_{h'} \sim \pi_{h'}(s_{h'}), s'_{h+1} \sim \mathbb{P}_h(\cdot \mid s'_h, a'_h) \right],$$

where the expectation is with respect to the randomness of the transition, the reward and the policy. Similarly, we can define the Q-value functions of policy $\pi$ by $\{Q_h^\pi\}_{h\in[H]} \subseteq (\mathcal{S} \times \mathcal{A} \to \mathbb{R})$ so that $Q_h^\pi(s, a)$ is equal to the expected cumulative reward an agent will receive if she follows policy $\pi$ starting from taking action $a$ at state $s$ and step $h$. Formally,

$$Q_h^\pi(s, a) = \mathbb{E}\left[ \sum_{h'=h}^H r_{h'}(s_{h'}, a_{h'}) \mid (s_h, a_h) = (s, a), a_{h'} \sim \pi_{h'}(s_{h'}), s'_{h+1} \sim \mathbb{P}_h(\cdot \mid s'_h, a'_h) \right].$$

We denote by $\pi^\star = \{\pi_h^\star\}_{h \in [H]}$ the optimal policy such that $\pi^\star \in \arg\max_\pi V_h^\pi(s)$ for all $(s, h) \in \mathcal{S} \times [H]$. By backward induction, one can easily prove that there always exists an optimal Markov policy. For simplicity of notations, we denote by $V_h^\star = V_h^{\pi^\star}$ and $Q_h^\star = Q_h^{\pi^\star}$ the optimal value functions. Note that given an MDP, the optimal value functions are unique despite that there may exist multiple optimal policies.

**Learning objective.** The objective of this paper is to design an efficient policy optimization algorithm to learn an $\epsilon$-optimal policy $\pi$ such that $V_1^\pi(s_1) \geq V_1^\star(s_1) - \epsilon$. Here the optimality is only measured by the value at the initial state $s_1$, because (a) each episode always starts from $s_1$, and (b) this paper studies the online exploration setting without access to a simulator, which means some states might be unreachable and learning optimal policies starting from those states is in general impossible.

# 3 Optimistic Natural Policy Gradient

In this section, we present the algorithm OPTIMISTIC NPG (Optimistic Natural Policy Gradient) and its theoretical guarantees.

## 3.1 Algorithm

The pseudocode of OPTIMISTIC NPG is provided in Algorithm 1. At a high level, the algorithm consists of the following three key modules.

- **Periodic on-policy data collection** (Lines 4-6): Similarly to the empirical PPO algorithm [Schulman et al., 2017] , OPTIMISTIC NPG discards all the old data, and executes the *current* policy $\pi^k$ to collect a batch of fresh data $\mathcal{D}^k$ after every $m$ steps of policy update. These data will later be used to evaluate and improve the policies in the next $m$ steps. Noticeably, this *on-policy* data mechanism is very different from most existing works in theoretical RL, where they either need to keep the historical data or have to rerun historical policies to refresh the dataset for the technical purpose of elliptical potential arguments in proofs. In comparison, OPTIMISTIC NPG only uses fresh data collected by the current (or very recent) policy, which resembles practical policy optimization algorithms such as PPO [Schulman et al., 2017] and TRPO [Schulman et al., 2015].

- **Optimistic policy evaluation** (Line 8): Given the above collected dataset $\mathcal{D}^k$, we estimate the value functions of the current policy $\pi^k$ by invoking Subroutine OPE (optimistic policy evaluation). In Section 4, we show how to implement Subroutine OPE for tabular MDPs, linear MDPs and RL problems of low eluder dimension.

- **Policy update** (Line 9): Given the optimistic value estimates $\{\overline{Q}_h^k\}_{h \in [H]}$ of $\pi^k$, OPTIMISTIC NPG performs one-step mirror ascent from $\pi_h^k(\cdot \mid s)$ with gradient $\overline{Q}_h^k(s, \cdot)$ to obtain $\pi_h^{k+1}(\cdot \mid s)$ at each $(h, s)$. Importantly, this step can be implemented in a computationally efficient way, because by the update rule of mirror ascent $\pi_h^k(\cdot \mid s) \propto \exp(\eta \sum_{t=1}^{k-1} \overline{Q}_h^t(s, \cdot))$, we only need to store $\{\{\overline{Q}_h^t\}_{h \in [H]}\}_{t \in [K]}$, from which any $\pi_h^k(\cdot \mid s)$ can be computed on the fly.

## 3.2 Theoretical guarantee

OPTIMISTIC NPG is a generic policy optimization meta-algorithm, and we can show it provably learns near-optimal policies as long as subroutine OPE satisfies the following condition.

**Condition 1** (Requirements for OPE). *Suppose parameter $m$ and $\eta$ in* OPTIMISTIC NPG *satisfy $m \leq (\eta H^2)^{-1}$. Then with probability at least $1 - \delta$,* OPE$(\pi^k, \mathcal{D}^k)$ *returns* $\{\overline{Q}_h^k\}_{h \in [H]} \subseteq (\mathcal{S} \times \mathcal{A} \to [0, H])$ *that satisfy the following properties for all $k \in [K]$.*

*(1A)* **Optimism:** *For all $(s, a, h) \in \mathcal{S} \times \mathcal{A} \times [H]$*

$$\overline{Q}_h^k(s, a) \geq (\mathcal{T}_h^{\pi^k} \overline{Q}_{h+1}^k)(s, a)$$

*where $(\mathcal{T}_h^\pi f)(s, a) = R_h(s, a) + \mathbb{E}[f(s', a') \mid s' \sim \mathbb{P}_h(\cdot \mid s, a), a' \sim \pi_{h+1}(s')]$.*

---

**Algorithm 1** OPTIMISTIC NPG

1: **input**: number of iterations $K$, period of collecting fresh data $m$, batch size $N$, learning rate $\eta$
2: **initialize:** for all $(h,s) \in [H] \times \mathcal{S}$ set $\pi_h^1(\cdot \mid s) = \text{Uniform}(\mathcal{A})$
3: **for** $k = 1, \ldots, K$ **do**
4:     **if** $k \equiv 1 (\text{mod } m)$ **then**
5:         $\mathcal{D}^k \leftarrow \{N \text{ fresh trajectories } \overset{\text{i.i.d.}}{\sim} \pi^k\}$
6:     **else**
7:         $\mathcal{D}^k \leftarrow \mathcal{D}^{k-1}$
8:     $\{\overline{Q}_h^k\}_{h \in [H]} \leftarrow \text{OPE}(\pi^k, \mathcal{D}^k)$
9:     for all $(h,s) \in [H] \times \mathcal{S}$ update $\pi_h^{k+1}(\cdot \mid s) \propto \pi_h^k(\cdot \mid s) \cdot \exp(\eta \cdot \overline{Q}_h^k(s, \cdot))$
10: **output**: $\pi^{\text{out}}$ that is sampled uniformly at random from $\{\pi^k\}_{k \in [K]}$

---

*(1B)* **Consistency:** *There exists an absolute complexity measure $L \in \mathbb{R}^+$ such that*

$$\overline{V}_1^k(s_1) - V_1^{\pi^k}(s_1) \leq \sqrt{(L/N) \times \log^2(NKL/\delta)},$$

*where $\overline{V}_1^k(s_1) = \mathbb{E}\left[\overline{Q}_1^k(s_1, a_1) \mid a_1 \sim \pi^k(s_1)\right].$*

Condition (1A) requires that the Q-value estimate returned by OPE satisfies the Bellman equation under policy $\pi^k$ optimistically. Requiring optimism is very common in analyzing RL algorithms, as most algorithms rely on the principle of optimism in face of uncertainty to handle exploration. Condition (1B) requires that the degree of over-optimism at initial state $s_1$ is roughly of order $\sqrt{1/N}$ with respect to the number of samples $N$. This is intuitive since as more data are collected from policy $\pi^k$ or some policy similar to $\pi^k$ (as is enforced by the precondition $m \leq (\eta H^2)^{-1}$), the value estimate at the initial state should be more accurate. In Section 4, we will provide concrete instantiations of Subroutine OPE for tabular MDPs, linear MDPs and RL problems of low eluder dimension, all of which satisfy Condition 1 with mild complexity measure $L$.

Under Condition 1, we have the following theoretical guarantees for OPTIMISTIC NPG, which we prove in Appendix A.

**Theorem 1.** *Suppose Condition 1 holds. In Algorithm 1, if we choose*

$$K = \Theta\left(\frac{H^4 \log |\mathcal{A}|}{\epsilon^2}\right), \quad N = \Theta\left(\frac{L \log^2(LK/\delta)}{\epsilon^2}\right), \quad \eta = \Theta\left(\frac{\epsilon}{H^3}\right), \qquad m \leq \Theta\left(\frac{H}{\epsilon}\right),$$

*then with probability at least $1/2$, $\pi^{\text{out}}$ is $\mathcal{O}(\epsilon)$-optimal.*

Below we emphasize two special choices of $m$ (the period of sampling fresh data) in Theorem 1:

- When choosing $m = 1$, OPTIMISTIC NPG is purely *on-policy* as it only uses data sampled from the current policy to perform policy optimization. In this case, the total sample complexity is

$$\frac{\text{\# iteration}}{\text{period of sampling}} \times \text{batch size} = \frac{K}{m} \times N = \tilde{\Theta}\left(\frac{LH^4}{\epsilon^4}\right),$$

  which, to our knowledge, is the first polynomial sample complexity guarantee for purely on-policy algorithms.

- When choosing $m = [H/\epsilon]$, we obtain sample complexity

$$\frac{\text{\# iteration}}{\text{period of sampling}} \times \text{batch size} = \frac{K}{m} \times N = \tilde{\Theta}\left(\frac{LH^3}{\epsilon^3}\right),$$

  which improves over the above purely on-policy version by a factor of $H/\epsilon$. In Section 4.2, we will specialize this result to linear MDPs to derive a sample complexity that improves the best existing one in policy optimization [Zanette et al., 2021] by a factor of $d$.

Finally we remark that for cleaner presentation, we state Theorem 1 for a fixed, constant failure probability. However, the result is easy to extend to the case when the failure probability is some arbitrary value of $\delta \in (0, 1)$ at the expense of increasing the sample complexity by a factor of $\log(1/\delta)$: one simply need to run Algorithm 1 for $\log(1/\delta)$ times, estimate the values of every output policy to accuracy $\epsilon$, and then pick the one with the highest estimate.

---

**Subroutine 1** Tabular OPE$(\pi, \mathcal{D})$

---

1: **required parameter**: $\alpha$
2: split $\mathcal{D}$ evenly into $H$ disjoint subsets $\mathcal{D}_1, \ldots, \mathcal{D}_H$
3: compute empirical estimate $(\hat{\mathbb{P}}_h, \hat{R}_h)$ of the transition and reward at step $h$ by using $\mathcal{D}_h$
4: set $\overline{V}_{H+1}(s) \leftarrow 0$ for all $s \in \mathcal{S}$
5: **for** $h = H : 1$ **do**
6:     for all $(s, a) \in \mathcal{S} \times \mathcal{A}$, compute
        $$J_h(s, a) \leftarrow \sum_{(s_h, a_h) \in \mathcal{D}_h} \mathbf{1}((s_h, a_h) = (s, a))$$
      and define
      $$\begin{cases} \overline{Q}_h(s, a) = \min\left\{ H - h + 1, \ \mathbb{E}_{s' \sim \hat{\mathbb{P}}_h(s,a)}[\overline{V}_{h+1}(s')] + \hat{R}_h(s, a) + b_h(s, a) \right\} \\ \overline{V}_h(s) = \mathbb{E}_{a \sim \pi_h(\cdot|s)}\left[\overline{Q}_h(s, a)\right] \end{cases}$$
      with bonus function
        $$b_h(s, a) = \alpha(J_h(s, a) + 1)^{-1/2}$$
7: **output** $\{\overline{Q}_h\}_{h=1}^H$

---

# 4 Examples

In this section, we implement subroutine OPE for tabular MDPs, linear MDPs and RL problems of low eluder dimension, and derive the respective sample complexity guarantees of OPTIMISTIC NPG.

## 4.1 Tabular MDPs

The implementation of tabular OPE (Subroutine 1) is rather standard: first estimate the transition and reward from dataset $\mathcal{D}$, then plug the estimated transition-reward into the Bellman equation under policy $\pi$ to compute the Q-value estimate backwards. And to guarantee optimism, we additionally add standard counter-based UCB bonus to compensate the error in model estimation. Formally, we have the following guarantee for tabular OPE.

**Proposition 1** (tabular MDPs). *Suppose we choose $\alpha = \Theta\left(H\sqrt{\log(KH|\mathcal{S}||\mathcal{A}|)}\right)$ in Subroutine 1, then Condition 1 holds with $L = SAH^3$.*

By combining Proposition 1 with Theorem 1, we prove that OPTIMISTIC NPG with Subroutine 1 learns an $\epsilon$-optimal policy within $(KN/m) = \tilde{\mathcal{O}}(H^6|\mathcal{S}||\mathcal{A}|/\epsilon^3)$ episodes for any tabular MDP. This rate is strictly worse than the best existing result $\tilde{\mathcal{O}}(H^3|\mathcal{S}||\mathcal{A}|/\epsilon^2)$ in tabular policy optimization [Wu et al., 2022]. Nonetheless, the proof techniques in [Wu et al., 2022] are specially tailored to tabular MDPs and it is highly unclear how to generalize them to linear MDPs due to certain technical difficulty that arises from covering a prohibitively large policy space. In comparison, our policy optimization framework easily extends to linear MDPs and provides improved sample complexity over the best existing one, as is shown in the following section.

## 4.2 Linear MDPs

We provide the instantiation of OPE for linear MDPs in Subroutine 2. At a high level, linear OPE computes an upper bound $\overline{Q}$ for $Q^\pi$ by using the Bellman equation under policy $\pi$ backwards from step $H$ to step 1 while adding a bonus to compensate the uncertainty of parameter estimation. Specifically, the step of ridge regression (Line 5) utilizes the linear completeness property of linear MDPs in computing $\overline{Q}_h$ from $\overline{V}_{h+1}$, which states that for any function $\overline{V}_{h+1} : \mathcal{S} \to \mathbb{R}$, there exists $\overline{\theta}_h \in \mathbb{R}^d$ so that $\langle \phi_h(s, a), \overline{\theta}_h \rangle = R_h(s, a) + \mathbb{E}_{s' \sim \mathbb{P}_h(\cdot|s,a)}[\overline{V}_{h+1}(s')]$ for all $(s, a) \in \mathcal{S} \times \mathcal{A}$. And in Line 6, we add an elliptical bonus $b_h(s, a)$ to compensate the error between our estimate $\hat{\theta}_h$ and the groundtruth $\overline{\theta}_h$ so that we can guarantee $\overline{Q}_h(s, a) \geq (\mathcal{T}_h^\pi \overline{Q}_{h+1})(s, a)$ for all $(s, a, h)$ with high probability .

$$\hat{\theta}_h = \text{argmin}_\theta \sum_{(s_h, a_h, r_h, s_{h+1}) \in \mathcal{D}_h} \left( \phi_h(s_h, a_h)^\mathrm{T}\theta - r_h - \overline{V}_{h+1}(s_{h+1}) \right)^2 + \lambda\|\theta\|_2^2. \tag{1}$$

---

**Subroutine 2** Linear OPE$(\pi, \mathcal{D})$

---

1: **required parameter**: $\alpha, \lambda$
2: split $\mathcal{D}$ evenly into $H$ disjoint subsets $\mathcal{D}_1, \ldots, \mathcal{D}_H$
3: set $\overline{V}_{H+1}(s) \leftarrow 0$ for all $s \in \mathcal{S}$
4: **for** $h = H : 1$ **do**
5:     perform ridge regression according to Equation (1)
6:     compute covariance matrix
$$\Sigma_h = \textstyle\sum_{(s_h, a_h) \in \mathcal{D}_h} \phi_h(s_h, a_h) \phi_h(s_h, a_h)^{\mathrm{T}}$$
    and define
$$\begin{cases} \overline{Q}_h(s, a) = \mathrm{Truncate}_{[0, H]}\big(\langle \hat{\theta}_h, \phi_h(s, a) \rangle + b_h(s, a)\big) \\ \overline{V}_h(s) = \mathbb{E}_{a \sim \pi_h(\cdot | s)} \big[\overline{Q}_h(s, a)\big] \end{cases}$$
    with bonus function
$$b_h(s, a) := \alpha \times \|\phi_h(s, a)\|_{(\Sigma_h + \lambda \mathbf{I}_{d \times d})^{-1}}$$
7: **output** $\{\overline{Q}_h\}_{h=1}^H$

---

---

**Subroutine 3** General OPE$(\pi, \mathcal{D})$

---

1: **required parameter**: $\beta$
2: split $\mathcal{D}$ evenly into $H$ disjoint subsets $\mathcal{D}_1, \ldots, \mathcal{D}_H$
3: set $\overline{V}_{H+1}(s) \leftarrow 0$ for all $s \in \mathcal{S}$
4: **for** $h = H : 1$ **do**
5:     **for** $(s, a) \in \mathcal{S} \times \mathcal{A}$ **do**
6:         construct confidence set
$$\mathcal{B}_h = \{f_h \in \mathcal{F}_h : \ \mathcal{L}_h(\mathcal{D}_h, f_h) \leq \min_{g_h \in \mathcal{F}_h} \mathcal{L}_h(\mathcal{D}_h, g_h) + \beta\}$$
        with loss function defined as
$$\mathcal{L}_h(\mathcal{D}_h, \zeta) = \sum_{(s_h, a_h, r_h, s_{h+1}) \in \mathcal{D}_h} \big(\zeta(s_h, a_h) - r_h - \overline{V}_{h+1}(s_{h+1})\big)^2$$
7:         compute
$$\begin{cases} \overline{Q}_h(s, a) = \sup_{f_h \in \mathcal{B}_h} f_h(s, a) \\ \overline{V}_h(s) = \mathbb{E}_{a \sim \pi_h(\cdot | s)} \big[\overline{Q}_h(s, a)\big] \end{cases}$$
8: **output** $\{\overline{Q}_h\}_{h=1}^H$

---

**Proposition 2** (linear MDPs). *Suppose we choose $\lambda = 1$ and $\alpha = \Theta\left(H\sqrt{d \log(KN)}\right)$ in Subroutine 2, then Condition 1 holds with $L = d^2 H^3$.*

By combining Proposition 2 with Theorem 1, we obtain that $(KN/m) = \tilde{\mathcal{O}}(d^2 H^6/\epsilon^3)$ episodes are sufficient for OPTIMISTIC NPG to learn an $\epsilon$-optimal policy, improving upon the state-of-the-art policy optimization results [Zanette et al., 2021] by a factor of $d$. Notably, this is also the first computationally efficient algorithm to achieve optimal quadratic dimension dependence for learning linear MDPs. The key factor behind this improvement is OPTIMISTIC NPG's periodic collection of fresh on-policy data, which eliminates the undesired correlation and avoids the union bound over certain nonlinear function class that are commonly observed in previous works. Consequently, OPTIMISTIC NPG uses a bonus function that is $\sqrt{d}$ times smaller than in previous works [e.g., Jin et al., 2020]. For further details on how we achieve this improvement, we refer interested readers to Appendix A, specifically Lemma 2.

### 4.3 General function approximation

Now we instantiate OPE for RL with general function approximation. In this setting, the learner is provided with a function class $\mathcal{F} = \mathcal{F}_1 \times \cdots \times \mathcal{F}_H$ for approximating the Q values of polices, where $\mathcal{F}_h \subseteq (\mathcal{S} \times \mathcal{A} \to [0, H])$.

**General OPE.** The pseudocode is provided in Subroutine 3. At each step $h \in [H]$, general OPE first constructs a confidence set $\mathcal{B}_h$ which contains candidate estimates for $\mathcal{T}_h^\pi \overline{Q}_{h+1}$ (Line 6). Specifically, $\mathcal{B}_h$ consists of all the value candidates $f_h \in \mathcal{F}_h$ whose square temporal difference (TD) error on dataset $\mathcal{D}_h$ is no larger than the smallest one by an additive factor $\beta$. Such construction can be viewed as a relaxation of the classic fitted Q-iteration (FQI) algorithm which only keeps the value candidate with the smallest TD error. In particular, if we pick $\beta = 0$, $\mathcal{B}_h$ collapses to the solution of FQI. Equipped with confidence set $\mathcal{B}_h$, general OPE then perform optimistic planning at every $(s, a) \in \mathcal{S} \times \mathcal{A}$ by computing $\overline{Q}_h(s, a)$ to be the largest possible value that can be achieved by any candidate in confidence set $\mathcal{B}_h$ (Line 7). We remark that general OPE shares a similar algorithmic spirit to the GOLF algorithm [Jin et al., 2021]. The key difference is that GOLF is a purely value-based algorithm and only performs optimistic planning at the initial state while general OPE performs optimistic planning at every state and is part of a policy gradient algorithm.

**Theoretical guarantee.** To state the main theorem, we need to first introduce two standard concepts: value closeness and eluder dimension, which have been widely used in previous works that study RL with general function approximation.

**Assumption 1** (value closeness [Wang et al., 2020]). *For all $h \in [H]$ and $V_{h+1} : \mathcal{S} \to [0, H]$, $\mathcal{T}_h V_{h+1} \in \mathcal{F}_h$ where $[\mathcal{T}_h V_{h+1}](s, a) = R_h(s, a) + \mathbb{E}[V_{h+1}(s') \mid s' \sim \mathbb{P}_h(\cdot \mid s, a)]$.*

Intuitively, Assumption 1 can be understood as requiring that the application of the Bellman operator $\mathcal{T}_h$ to any V-value function $V_{h+1}$ results in a function that belongs to the value function class $\mathcal{F}_h$. This assumption holds in various settings, including tabular MDPs and linear MDPs. Furthermore, it is reasonable to assume that value closeness holds whenever the function class $\mathcal{F}$ has sufficient expressive power, such as the class of neural networks.

**Definition 1** (eluder dimension [Russo and Van Roy, 2013]). *Let $\mathcal{G}$ be a function class from $\mathcal{X}$ to $\mathbb{R}$ and $\epsilon > 0$. We define the $\epsilon$-eluder dimension of $\mathcal{G}$, denoted as $d_E(\mathcal{G}, \epsilon)$, to be the largest $L \in \mathbb{N}^+$ such that there exists $x_1, \ldots, x_L \in \mathcal{X}$ and $g_1, g_1', \ldots, g_L, g_L' \in \mathcal{G}$ satisfying: for all $l \in [L]$, $\sum_{i<l}(g_l(x_i) - g_l'(x_i))^2 \le \epsilon$ but $g_l(x_l) - g_l'(x_l) \ge \epsilon$.*

At a high level, eluder dimension $d_E(\mathcal{G}, \epsilon)$ measures how many mistakes we have to make in order to identify an unknown function from function class $\mathcal{G}$ to accuracy $\epsilon$, in the worst case. It has been widely used as a sufficient condition for proving sample-efficiency guarantee for optimistic algorithms in RL with general function approximation, e.g., Wang et al. [2020], Jin et al. [2021].

Now we state the theoretical guarantee for general OPE.

**Proposition 3** (general function approximation). *Suppose Assumption 1 holds and we choose $\beta = \Theta\left(H^2 \log(|\mathcal{F}| NK/\delta)\right)$ in Subroutine 3, then Condition 1 holds with $L = H^3 \log(|\mathcal{F}|) \max_h d_E(\mathcal{F}_h, 1/N)$.*

Plugging Proposition 3 back into Theorem 1, we obtain that $(KN/m) = \tilde{\mathcal{O}}(d_E \log(|\mathcal{F}|)H^6/\epsilon^3)$ episodes are sufficient for OPTIMISTIC NPG to learn an $\epsilon$-optimal policy, which to our knowledge is the first polynomial sample complexity guarantee for policy optimization with general function approximation.

## 5   Conclusions

We proposed a model-free, policy optimization algorithm—OPTIMISTIC NPG for online RL and analyzed its behavior in the episodic setting. In terms of algorithmic design, it is not only considerably simpler but also more closely resembles the empirical policy optimization algorithms (e.g., PPO, TRPO) than the previous theoretical algorithms. In terms of sample efficiency, for $d$-dimensional linear MDPs, it improves over state-of-the-art policy optimization algorithm by a factor of $d$, and is the first computationally efficient algorithm to achieve the optimal dimension dependence. To our best knowledge, OPTIMISTIC NPG is also the first sample-efficient policy optimization algorithm under general function approximation.

For future research, we believe that it is an important direction to investigate the optimal complexity for using policy optimization algorithms to solve linear MDPs. Despite the optimal dependence on dimension $d$, the current sample complexity of OPTIMISTIC NPG is worse than the optimal rate by a factor of $1/\epsilon$. It remains unclear to us how to shave off this $1/\epsilon$ factor to achieve the optimal rate.

## Acknowledgement

Qinghua Liu would like to thank Yu Bai for valuable discussions. This work was partially supported by National Science Foundation Grant NSF-IIS-2107304.

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

# A Proof of Theorem 1

Recall in Algorithm 1, $\pi^{\text{out}}$ is sampled uniformly at random from $\{\pi^k\}_{k\in[K]}$, so we have

$$\mathbb{E}\left[V_1^\star(s_1) - V_1^{\pi^{\text{out}}}(s_1)\right]$$

$$= \frac{1}{K}\sum_{k=1}^{K}\left[V_1^\star(s_1) - V_1^{\pi^k}(s_1)\right]$$

$$= \underbrace{\frac{1}{K}\sum_{k=1}^{K}\left[V_1^\star(s_1) - \overline{V}_1^k(s_1)\right]}_{\textbf{Term (I)}} + \underbrace{\frac{1}{K}\sum_{k=1}^{K}\left[\overline{V}_1^k(s_1) - V_1^{\pi^k}(s_1)\right]}_{\textbf{Term (II)}}.$$

To control Term (I), we import the following generalized policy difference lemma from [Shani et al., 2020, Cai et al., 2020].

**Lemma 1** (generalized policy-difference lemma). *For any policy $\pi$ and $k \in [K]$,*

$$V_1^\pi(s_1) - \overline{V}_1^k(s_1)$$

$$= \sum_{h=1}^{H}\mathbb{E}_{s_h\sim\pi}\left[\langle\pi_h(\cdot\mid s_h) - \pi_h^k(\cdot\mid s_h), \overline{Q}_h^k(s_h,\cdot)\rangle\right]$$

$$- \sum_{h=1}^{H}\mathbb{E}_{(s_h,a_h)\sim\pi}\left[\overline{Q}_h^k(s_h,a_h) - (\mathcal{T}_h^{\pi^k}\overline{Q}_{h+1}^k)(s_h,a_h)\right].$$

By invoking Lemma 1 with $\pi = \pi^\star$, we have

$$\text{Term(I)} = \frac{1}{K}\sum_{k=1}^{K}\left[V_1^\star(s_1) - \overline{V}_1^k(s_1)\right]$$

$$\leq H\max_{h,s}\left[\frac{1}{K}\sum_{k=1}^{K}\langle\pi_h^\star(\cdot\mid s) - \pi_h^k(\cdot\mid s), \overline{Q}_h^k(s,\cdot)\rangle\right] + H\max_{h,s,a}\left[(\mathcal{T}_h^{\pi^k}\overline{Q}_{h+1}^k - \overline{Q}_h^k)(s,a)\right] \quad (2)$$

$$\leq \mathcal{O}\left(\frac{H\log|\mathcal{A}|}{\eta K} + \eta H^3\right) + H\max_{h,s,a}\left[(\mathcal{T}_h^{\pi^k}\overline{Q}_{h+1}^k - \overline{Q}_h^k)(s,a)\right],$$

where the second inequality follows from the standard regret bound of online mirror ascent. By Condition (1A), the second term on the RHS of Equation (2) is upper bounded by zero. So we have

$$\text{Term(I)} \leq \mathcal{O}\left(\frac{H\log|\mathcal{A}|}{\eta K} + \eta H^3\right).$$

As for Term (II), we can directly upper bound it by Condition (1B) which gives:

$$\text{Term(II)} \leq H\sqrt{(L/N)\times\log(NKL/\delta)}.$$

We complete the proof by plugging in the choice of $K$, $N$, $\eta$.

# B Proofs for Examples

In this section, we prove the three instantiations of OPE in Section 4 satisfy Condition 1 with mild complexity measure $L$. The proofs of the lemmas used in this section can be found in Appendix C.

## B.1 Proof of Proposition 2 (linear OPE)

We start with the following lemma showing that $\overline{Q}_h^k$ is an optimistic estimate of $\mathcal{T}_h^{\pi^k}\overline{Q}_{h+1}^k$ and the degree of over-optimism can be bounded by the bonus function.

**Lemma 2** (optimism of value estimates). *Under the same choice of $\lambda$ and $\alpha$ as in Proposition 2, with probability at least $1 - \delta$: for all $(k, h, s, a) \in [K] \times [H] \times \mathcal{S} \times \mathcal{A}$:*

$$0 \leq (\overline{Q}_h^k - \mathcal{T}_h^{\pi^k} \overline{Q}_{h+1}^k)(s, a) \leq 2b_h^k(s, a),$$

*where $\{b_h^k\}_{h \in [H]}$ are the bonus functions defined in Line 6, Subroutine 2 linear $\mathrm{OPE}(\pi^k, \mathcal{D}^k)$.*

Importantly, Lemma 2 holds with a bonus function that is smaller by a factor of $\sqrt{d}$ than the typical one used in linear function approximation [e.g., Jin et al., 2020], which is key to obtaining the optimal dimension dependence. The main reason that this smaller bonus function suffices is that we recollect fresh samples every $m$ iterations and split the data (Line 2 in Subroutine 2) to eliminate the correlation between different steps $h \in [H]$.

Lemma 2 immediately implies that Condition (1A) holds. Below we prove Condition (1B).

By the second inequality in Lemma 2, the definition of $\overline{V}^k$ and the Bellman equation under policy $\pi^k$, one can show

$$\overline{V}_1^k(s_1) - V_1^{\pi^k}(s_1) = \sum_{h=1}^H \mathbb{E}_{\pi^k}[(\overline{Q}_h^k - \mathcal{T}_h^{\pi^k} \overline{Q}_{h+1}^k)(s_h, a_h)] \leq 2 \sum_{h=1}^H \mathbb{E}_{\pi^k}[b_h^k(s_h, a_h)]. \tag{3}$$

Denote by $t_k$ the index of the last iteration of collecting fresh data before the $k^{\text{th}}$ iteration. By exploiting the softmax parameterization structure of $\{\pi^k\}_{k \in [K]}$, one can show that the visitation measure over state-action pairs induced by executing policy $\pi^k$ is actually very close to the one induced by $\pi^{t_k}$ in the following sense.

**Lemma 3** (policy Lipschitz). *Suppose we choose $\eta$ and $m$ such that $\eta m \leq 1/H^2$, then for any $k \in \mathbb{N}^+$ and any function $f : \mathcal{S} \times \mathcal{A} \to \mathbb{R}^+$:*

$$\mathbb{E}_{\pi^k}[f(s_h, a_h)] = \Theta\left(\mathbb{E}_{\pi^{t_k}}[f(s_h, a_h)]\right).$$

By combining Lemma 3 with Equation (3) and noticing that $b_h^k = b_h^{t_k}$, we have

$$\overline{V}_1^k(s_1) - V_1^{\pi^k}(s_1) \leq \mathcal{O}\left(\max_{k \in [K]} \sum_{h=1}^H \mathbb{E}_{\pi^{t_k}}[b_h^k(s_h, a_h)]\right) = \mathcal{O}\left(\max_{k \in [K]} \sum_{h=1}^H \mathbb{E}_{\pi^{t_k}}[b_h^{t_k}(s_h, a_h)]\right).$$

Since the bonus function $b_h^{t_k}$ is defined by using the dataset $\mathcal{D}^{t_k}$ collected from executing policy $\pi^{t_k}$, one could expect its average under $\pi^{t_k}$ should be rather small when the batch size $N$ is large enough, as formalized in the following lemma.

**Lemma 4** (on-policy uncertainty). *Let $\pi$ be an arbitrary policy and $\lambda \geq 1$. Suppose we sample $\{(s_h^n, a_h^n)\}_{n=1}^N$ i.i.d. from $\pi$. Denote $\Sigma_h = \sum_{i=1}^N \phi_h(s_h^i, a_h^i)\phi_h^{\mathrm{T}}(s_h^i, a_h^i)$. Then with probability at least $1 - \delta$:*

$$\mathbb{E}_{(s_h, a_h) \sim \pi}\left[\|\phi_h(s_h, a_h)\|_{(\Sigma_h + \lambda \mathbf{I}_{d \times d})^{-1}}\right] = \mathcal{O}\left(\sqrt{\frac{d \log(N/\delta)}{N}}\right).$$

As a result, by Lemma 4 and the definition of $b_h^k$, we have that with probability at least $1 - \delta$: for all $k \in [K]$,

$$\sum_{h=1}^H \mathbb{E}_{\pi^{t_k}}[b_h^k(s_h, a_h)] = \mathcal{O}\left(Hd\sqrt{\frac{H \log^2(NKH/\delta)}{N}}\right)$$

where in applying Lemma 4 we use the fact $|\mathcal{D}_h^k| = N/H$ due to data splitting. Putting all pieces together, we get

$$\overline{V}_1^k(s_1) - V_1^{\pi^k}(s_1) \leq \mathcal{O}\left(\sqrt{\frac{d^2 H^3 \log^2(KNH/\delta)}{N}}\right).$$

As a result, Condition (1B) holds with $L = d^2 H^3$.

## B.2 Proof of Proposition 1 (tabular OPE)

The proof follows basically the same as that of Proposition 2. The only modification needed is to replace Lemma 2 and 4 by their tabular counterparts, which we state below.

**Lemma 5** (optimism of value estimates: tabular). *Under the same choice of $\alpha$ as in Proposition 1, with probability at least $1 - \delta$: for all $(k, h, s, a) \in [K] \times [H] \times \mathcal{S} \times \mathcal{A}$:*

$$0 \leq (\overline{Q}_h^k - \mathcal{T}_h^{\pi^k} \overline{Q}_{h+1}^k)(s,a) \leq 2b_h^k(s,a),$$

*where $\{b_h^k\}_{h \in [H]}$ are the bonus functions defined in Line 6, Subroutine 1 tabular $\mathrm{OPE}(\pi^k, \mathcal{D}^k)$.*

**Lemma 6** (on-policy uncertainty: tabular). *Let $\pi$ be an arbitrary policy. Suppose we sample $\{(s_h^n, a_h^n)\}_{n=1}^N$ i.i.d. from $\pi$. Denote $J_h(s,a) = \sum_{i=1}^N \mathbf{1}((s_h^i, a_h^i) = (s,a))$. Then with probability at least $1 - \delta$:*

$$\mathbb{E}_{(s_h, a_h) \sim \pi} \left[ \sqrt{\frac{1}{J_h(s,a) + 1}} \right] = \mathcal{O}\left( \sqrt{\frac{SA \log(N/\delta)}{N}} \right).$$

## B.3 Proof of Proposition 3 (general OPE)

The proof follows basically the same as that of Proposition 2. The only modification needed is to replace Lemma 2 and 4 by their general counterparts, which we state below.

**Lemma 7** (optimism of value estimates: general). *Suppose Assumption 1 holds. There exists an absolute constant $c$ such that with probability at least $1 - \delta$: for all $(k, h, s, a) \in [K] \times [H] \times \mathcal{S} \times \mathcal{A}$:*

$$0 \leq (\overline{Q}_h^k - \mathcal{T}_h^{\pi^k} \overline{Q}_{h+1}^k)(s,a) \leq b_h(s,a,\mathcal{D}_h^k) := \left( \begin{array}{c} \sup\limits_{f_h, f_h' \in \mathcal{F}_h} f_h(s,a) - f_h'(s,a) \\ s.t. \sum\limits_{(s_h,a_h) \in \mathcal{D}_h^k} (f_h(s_h,a_h) - f_h'(s_h,a_h))^2 \leq c\beta \end{array} \right).$$

**Lemma 8** (on-policy uncertainty: general). *Let $\pi$ be an arbitrary policy. Suppose we sample $\mathcal{D}_h = \{(s_h^n, a_h^n)\}_{n=1}^N$ i.i.d. from $\pi$. Then with probability at least $1 - \delta$:*

$$\mathbb{E}_{(s_h,a_h) \sim \pi} [b_h(s,a,\mathcal{D}_h)] = \mathcal{O}\left( H\sqrt{\frac{d_E(1/N, \mathcal{F}_h)\beta}{N}} + H\sqrt{\frac{\log(1/\delta)}{N}} \right).$$

# C Proofs of Lemmas

## C.1 Proof of Lemma 1

Lemma 1 is an immediate consequence of Lemma 4.2 in [Cai et al., 2020] or Lemma 1 in [Shani et al., 2020].

## C.2 Proof of Lemma 2

Let us consider a fixed pair $(k, h) \in [K] \times [H]$. Recall $\overline{Q}_h^k$ is defined as

$$\overline{Q}_h^k(s,a) = \mathrm{Truncate}_{[0,H-h+1]}\left( \langle \hat{\theta}_h^k, \phi_h(s,a) \rangle + b_h^k(s,a) \right),$$

where

$$\hat{\theta}_h^k = \mathrm{argmin}_\theta \sum_{(s_h, a_h, r_h, s_{h+1}) \in \mathcal{D}_h^k} \left( \phi_h(s_h, a_h)^{\mathrm{T}} \theta - r_h - \overline{V}_{h+1}^k(s_{h+1}) \right)^2 + \lambda \|\theta\|_2^2,$$

$$b_h^k(s,a) = \alpha \times \|\phi_h(s,a)\|_{(\Sigma_h^k + \lambda \mathbf{I}_{d \times d})^{-1}}, \quad \text{and} \quad \Sigma_h^k = \sum_{(s_h, a_h) \in \mathcal{D}_h^k} \phi(s_h, a_h)(\phi(s_h, a_h))^{\mathrm{T}}.$$

By the standard linear completeness property of linear MDPs Jin et al. [2020], there exists $\theta_h^k$ such that $\|\theta_h^k\|_2 \leq H\sqrt{d}$ and

$$\langle \phi_h(s,a), \theta_h^k \rangle = R_h(s,a) + \mathbb{E}_{s' \sim \mathbb{P}(\cdot|s,a)}\left[ \overline{V}_{h+1}^k(s') \right] \quad \text{for all } (s,a) \in \mathcal{S} \times \mathcal{A}.$$

Therefore, to prove Lemma 2, it suffices to show that for all $(s, a)$

$$\left| \langle \phi_h(s, a), \theta_h^k - \hat{\theta}_h^k \rangle \right| \leq b_h^k(s, a).$$

To condense notations, denote by $\{(s_h^i, a_h^i, r_h^i, s_{h+1}^i)\}_{i=1}^M$ the tuples in $\mathcal{D}_h^k$. By the definition of ridge regression,

$$\left| \langle \phi_h(s, a), \hat{\theta}_h^k - \theta_h^k \rangle \right|$$

$$= \left| \left\langle \phi_h(s, a) , (\Sigma_h^k + \lambda \mathbf{I}_{d \times d})^{-1} \times \right. \right.$$

$$\left. \left. \left( \sum_{i=1}^M \phi_h(s_h^i, a_h^i) \left( r_h^i + \overline{V}_{h+1}^k(s_{h+1}^i) - \phi_h(s_h^i, a_h^i)^\mathrm{T} \theta_h^k \right) - \lambda \theta_h^k \right) \right\rangle \right|$$

$$\leq \| \phi_h(s, a) \|_{(\Sigma_h^k + \lambda \mathbf{I}_{d \times d})^{-1}} \times$$

$$\left( \left\| \sum_{i=1}^M \phi_h(s_h^i, a_h^i) \left( r_h^i + \overline{V}_{h+1}^k(s_{h+1}^i) - \phi_h(s_h^i, a_h^i)^\mathrm{T} \theta_h^k \right) \right\|_{(\Sigma_h^k + \lambda \mathbf{I}_{d \times d})^{-1}} + H \sqrt{d} \right),$$

where the inequality follows from $\| \lambda \theta_h^k \|_{(\Sigma_h^k + \lambda \mathbf{I}_{d \times d})^{-1}} \leq \sqrt{\lambda} \| \theta_h^k \|_2 \leq H \sqrt{d}$. It remains to bound the first term in the bracket. Now here comes the key observation that helps shave off the $\sqrt{d}$ factor from the bonus.

**Observation 1.** *$\mathcal{D}_h^k$ and $\overline{V}_{h+1}^k$ are independent conditioning on $\pi^{t_k}$ where $t_k$ denotes the index of the last iteration of collecting fresh data before the $k^{\mathrm{th}}$ iteration.*

To see why, recall that after obtaining $\mathcal{D}^{t_k}$ Algorithm 2 splits it evenly into $H$ disjoint subsets $\{\mathcal{D}_h^{t_k}\}_{h=1}^H$ and then only uses $\{\mathcal{D}_{h'}^{t_k}\}_{h'=h+1}^H$ for evaluating and improving $\{\pi_{h'}^l\}_{h'=h+1}^H$ for $l \in [t_k, t_k + m - 1]$. As a result, $\{\overline{V}_{h+1}^l\}_{l=t_k}^{t_k+m-1}$ (including $\overline{V}_{h+1}^k$) is independent of $\mathcal{D}_h^{t_k} = \mathcal{D}_h^k$ conditioning on $\pi^{t_k}$. By the concentration of self-normalized processes [Abbasi-Yadkori et al., 2011], we conclude that with probability at least $1 - \delta$: for all $(k, h) \in [K] \times [H]$

$$\left\| \sum_{i=1}^M \phi_h(s_h^i, a_h^i) \left( r_h^i + \overline{V}_{h+1}^k(s_{h+1}^i) - \phi_h(s_h^i, a_h^i)^\mathrm{T} \theta_h^k \right) \right\|_{(\Sigma_h^k + \lambda \mathbf{I}_{d \times d})^{-1}} \leq \mathcal{O} \left( H \sqrt{d \log(MKH/\delta)} \right).$$

### C.3 Proof of Lemma 3

We first introduce the following two auxiliary lemmas about the lipschitzness continuity of softmax paraterization.

**Lemma 9.** *There exists an absolute constant $c > 0$ such that for any policy $\pi, \hat{\pi}$ satisfying $\hat{\pi}_h(a \mid s) \propto \pi_h(a \mid s) \times \exp(L_h(s, a))$ where $\{L_h(s, a)\}_{h \in [H]}$ is set of functions from $\mathcal{S} \times \mathcal{A}$ to $[-1/H, 1/H]$, we have that for any $\tau_H := (s_1, a_1, \ldots, s_H, a_H) \in (\mathcal{S} \times \mathcal{A})^H$:*

$$\mathbb{P}^{\hat{\pi}}(\tau_H) \leq c \times \mathbb{P}^{\pi}(\tau_H).$$

*Proof of Lemma 9.* By using the relation between $\pi$ and $\hat{\pi}$ and the normalization property of policy $\pi$, we have that for any $(h, s, a)$:

$$\hat{\pi}_h(a \mid s) = \frac{\pi_h(a \mid s) \times \exp(L_h(s, a))}{\sum_{a'} \pi_h(a' \mid s) \times \exp(L_h(s, a'))}$$

$$\leq \frac{\pi_h(a \mid s) \times \exp(1/H)}{\sum_{a'} \pi_h(a' \mid s) \exp(-1/H)}$$

$$= \pi_h(a \mid s) \times \exp(2/H) \leq \pi_h(a \mid s) \times \left( 1 + \frac{\mathcal{O}(1)}{H} \right).$$

Therefore, for any $\tau_H := (s_1, a_1, \ldots, s_H, a_H)$

$$\mathbb{P}^{\hat{\pi}}(\tau_H) = \left(\prod_{h=1}^{H-1} \mathbb{P}_h(s_{h+1} \mid s_h, a_h)\right) \times \left(\prod_{h=1}^{H} \hat{\pi}_h(a_h \mid s_h)\right)$$

$$\leq \left(\prod_{h=1}^{H-1} \mathbb{P}_h(s_{h+1} \mid s_h, a_h)\right) \times \left(\prod_{h=1}^{H} \pi_h(a_h \mid s_h)\right) \times \left(1 + \frac{\mathcal{O}(1)}{H}\right)^H$$

$$=\mathcal{O}(1) \times \left(\prod_{h=1}^{H-1} \mathbb{P}_h(s_{h+1} \mid s_h, a_h)\right) \times \left(\prod_{h=1}^{H} \pi_h(a_h \mid s_h)\right) = \mathcal{O}(1) \times \mathbb{P}^{\pi}(\tau_H).$$

$\square$

**Lemma 10.** *Let $\mu, \nu$ be two probability densities defined over $\mathcal{X}$ such that $\|\mu/\nu\|_\infty \leq \alpha$, then we have that for any function $f : \mathcal{X} \to \mathbb{R}^+$, $\mathbb{E}_\mu[f(x)] \leq \alpha \mathbb{E}_\nu[f(x)]$.*

*Proof of Lemma 10.* By definition, $\mathbb{E}_\mu[f(x)] = \int_x f(x)\mu(x)dx \leq \int_x f(x)(\alpha\nu(x))dx = \alpha\mathbb{E}_\nu[f(x)]$. $\square$

By the update rule of NPG, we have

$$\pi_h^k(\cdot \mid s) \propto \pi_h^{t_k}(\cdot \mid s) \times \exp\left(\eta \sum_{i=t_k}^{k-1} \overline{Q}_h^i(s, \cdot)\right).$$

Since we choose $\eta$ and $m$ such that $\eta m \leq 1/H^2$,

$$\left|\eta \sum_{i=t_k}^{k-1} \overline{Q}_h^i(s, \cdot)\right| \leq \eta(k - t_k)H \leq \eta m H \leq 1/H.$$

Therefore, by invoking Lemma 9 with $\hat{\pi} = \pi^k$ and $\pi = \pi^{t_k}$, we have that for any $\tau_H \in (\mathcal{S} \times \mathcal{A})^H$:

$$\mathbb{P}^{\pi^k}(\tau_H) \leq c \times \mathbb{P}^{\pi^{t_k}}(\tau_H).$$

By further invoking Lemma 10 with $\mathcal{X} = (\mathcal{S} \times \mathcal{A})^H$, $\mu = \mathbb{P}^{\pi^k}$ and $\nu = \mathbb{P}^{\pi^{t_k}}$, we obtain that for any function $f : \mathcal{S} \times \mathcal{A} \to \mathbb{R}^+$:

$$\mathbb{E}_{\pi^k}[f(s_h, a_h)] = \mathcal{O}\left(\mathbb{E}_{\pi^{t_k}}[f(s_h, a_h)]\right).$$

Similarly, we can show $\mathbb{E}_{\pi^{t_k}}[f(s_h, a_h)] = \mathcal{O}\left(\mathbb{E}_{\pi^k}[f(s_h, a_h)]\right)$, which completes the proof of Lemma 3.

### C.4 Proof of Lemma 4

To simplify notations, denote $\phi_h^i := \phi_h(s_h^i, a_h^i)$. For the technical purpose of applying martingale concentration, we additionally define the intermediate empirical covariance matrices: $\Sigma_h^n = \sum_{i=1}^{n-1} \phi_h^i(\phi_h^i)^{\mathrm{T}}$, for $n \in [N]$. We have that with probability at least $1 - \delta$,

$$\mathbb{E}_{(s_h, a_h)\sim\pi}\left[\|\phi_h(s_h, a_h)\|_{(\Sigma_h + \lambda\mathbf{I}_{d\times d})^{-1}}\right]$$

$$\leq \frac{1}{N}\sum_{n=1}^{N} \mathbb{E}_{(s_h, a_h)\sim\pi}\left[\|\phi_h(s_h, a_h)\|_{(\Sigma_h^n + \lambda\mathbf{I}_{d\times d})^{-1}}\right]$$

$$\leq \frac{1}{N}\sum_{n=1}^{N} \|\phi_h^n\|_{(\Sigma_h^n + \lambda\mathbf{I}_{d\times d})^{-1}} + \mathcal{O}\left(\sqrt{\frac{\log(1/\delta)}{N}}\right) \tag{4}$$

$$\leq \mathcal{O}\left(\sqrt{\frac{d\log N}{N}}\right) + \mathcal{O}\left(\sqrt{\frac{\log(1/\delta)}{N}}\right),$$

where the first inequality uses $\Sigma_h^n \preceq \Sigma_h$, the second one uses Azuma-Hoeffding inequality with $\lambda = 1$, and the last one uses Cauchy–Schwarz inequality and the the standard elliptical potential argument.

## C.5 Proof of Lemma 5

Let us consider a fixed pair $(k, h) \in [K] \times [H]$. Recall $\overline{Q}_h^k$ is defined as

$$\overline{Q}_h^k(s, a) = \min \left\{ H - h + 1, \ \mathbb{E}_{s' \sim \hat{\mathbb{P}}_h^k(s,a)}[\overline{V}_{h+1}^k(s')] + \hat{R}_h^k(s, a) + b_h^k(s, a) \right\}$$

where $\hat{\mathbb{P}}_h^k$ and $\hat{\mathbb{R}}_h^k$ are the empirical estimates of transition and reward at step $h$ from $\mathcal{D}_h^k$, and $b_h^k(s, a) = \alpha(J_h^k(s, a) + 1)^{-1/2}$ with $J_h^k(s, a) = \sum_{(s_h, a_h) \in \mathcal{D}_h^k} \mathbf{1}((s_h, a_h) = (s, a))$. Therefore, to prove Lemma 5, it suffices to show that for all $(s, a)$

$$\left| \left( \mathbb{E}_{s' \sim \hat{\mathbb{P}}_h^k(s,a)}[\overline{V}_{h+1}^k(s')] - \mathbb{E}_{s' \sim \mathbb{P}_h(s,a)}[\overline{V}_{h+1}^k(s')] \right) + \left( \hat{R}_h^k(s, a) - R_h(s, a) \right) \right| \leq b_h^k(s, a).$$

By observation 1, we know $\{\overline{V}_{h+1}^l\}_{l=t_k}^{t_k+m-1}$ is independent of $\mathcal{D}_h^{t_k} = \mathcal{D}_h^k$ conditioning on $\pi^{t_k}$, which implies $\overline{V}_{h+1}^k$ is independent of $\hat{\mathbb{P}}_h^k$. Therefore, by Azuma-Hoeffding inequality and standard union bound, we conclude that with probability at least $1 - \delta$: for all $(k, h, s, a) \in [K] \times [H] \times \mathcal{S} \times \mathcal{A}$,

$$\left| \left( \mathbb{E}_{s' \sim \hat{\mathbb{P}}_h^k(s,a)}[\overline{V}_{h+1}^k(s')] - \mathbb{E}_{s' \sim \mathbb{P}_h(s,a)}[\overline{V}_{h+1}^k(s')] \right) + \left( \hat{R}_h^k(s, a) - R_h(s, a) \right) \right|$$
$$\leq \mathcal{O} \left( H \sqrt{\frac{\log(KNHSA/\delta)}{J_h^k(s, a) + 1}} \right).$$

## C.6 Proof of Lemma 6

For the technical purpose of performing martingale concentration, we introduce the notion of intermediate counters: $J_h^n(s, a) = \sum_{i=1}^{n-1} \mathbf{1}((s_h^i, a_h^i) = (s, a))$, for $n \in [N]$. We have that with probability at least $1 - \delta$,

$$\mathbb{E}_{(s_h, a_h) \sim \pi} \left[ \sqrt{\frac{1}{J_h(s, a) + 1}} \right]$$
$$\leq \frac{1}{N} \sum_{n=1}^N \mathbb{E}_{(s_h, a_h) \sim \pi} \left[ \sqrt{\frac{1}{J_h^n(s, a) + 1}} \right] \tag{5}$$
$$\leq \frac{1}{N} \sum_{n=1}^N \sqrt{\frac{1}{J_h^n(s_h^n, a_h^n) + 1}} + \mathcal{O} \left( \sqrt{\frac{\log(1/\delta)}{N}} \right)$$
$$\leq \mathcal{O} \left( \sqrt{\frac{SA \log N}{N}} \right) + \mathcal{O} \left( \sqrt{\frac{\log(1/\delta)}{N}} \right),$$

where the first inequality uses $J_h^n(s, a) \leq J_h(s, a)$ for all $(s, a) \in \mathcal{S} \times \mathcal{A}$, the second one uses Azuma-Hoeffding inequality, and the last one follows the standard pigeon-hole argument.

## C.7 Proof of Lemma 7

By observation 1, we know $\mathcal{D}_h^k$ is independent of $\overline{V}_{h+1}^k$ conditioning on $\pi^{t_k}$ where $t_k$ is the index of the last iteration when OPTIMISTIC NPG collects fresh data before iteration $k + 1$. With this independence relation in mind, we can easily prove that confidence set $\mathcal{B}_h^k$ satisfy the following properties by standard concentration inequality and union bounds.

**Lemma 11.** *Suppose Assumption 1 holds. There exists an absolute constant $c$ such that with probability at least $1 - \delta$, for all $k \in [K]$ and $h \in [H]$,*

- $\mathcal{T}_h^{\pi^k} \overline{Q}_{h+1}^k \in \mathcal{B}_h^k$,

- *for any $f_h \in \mathcal{B}_h^k$, $\sum_{(s_h, a_h) \in \mathcal{D}_h^k} (f_h(s_h, a_h) - \mathcal{T}_h^{\pi^k} \overline{Q}_{h+1}^k(s_h, a_h))^2 \leq c\beta$.*

The proof of Lemma 11 follows trivially from modifying the proofs of Lemma 39 and 40 in Jin et al. [2021], which we omit here.

Define
$$\bar{\mathcal{B}}_h^k = \{f_h \in \mathcal{F}_h : \sum_{(s_h,a_h) \in \mathcal{D}_h^k} (f_h(s_h,a_h) - (\mathcal{T}_h^{\pi^k}\overline{Q}_{h+1}^k)(s_h,a_h))^2 \leq c\beta\}.$$

By using the first relation in Lemma 11 and the definition of $\overline{Q}_h^k$, we immediately obtain that
$$(\mathcal{T}_h^{\pi^k}\overline{Q}_{h+1}^k)(s,a) \leq \sup_{f_h \in \mathcal{B}_h^k} f_h(s,a) = \overline{Q}_h^k(s,a)$$

and
$$\overline{Q}_h^k(s,a) - (\mathcal{T}_h^{\pi^k}\overline{Q}_{h+1}^k)(s,a) \leq \sup_{f_h,f_h' \in \mathcal{B}_h^k} (f_h(s,a) - f_h'(s,a)) \leq \sup_{f_h,f_h' \in \bar{\mathcal{B}}_h^k} (f_h(s,a) - f_h'(s,a)),$$

where the last inequality uses $\bar{\mathcal{B}}_h^k \subset \mathcal{B}_h^k$ by Lemma 11.

### C.8 Proof of Lemma 8

Recall we define
$$b_h(s,a,\mathcal{D}_h) := \left( \begin{array}{c} \sup\limits_{f_h,f_h' \in \mathcal{F}_h} f_h(s,a) - f_h'(s,a) \\ \text{s.t.} \sum\limits_{(s_h,a_h) \in \mathcal{D}_h} (f_h(s_h,a_h) - f_h'(s_h,a_h))^2 \leq c\beta \end{array} \right),$$

which implies that $b_h(s,a,\mathcal{D}_h) \leq b_h(s,a,\underline{\mathcal{D}}_h)$ for any $\underline{\mathcal{D}}_h \subseteq \mathcal{D}_h$. Let $\mathcal{D}_h^{(n)} = \{(s_h^i, a_h^i)\}_{i=1}^n$. We have that with probability at least $1 - \delta$,

$$\mathbb{E}_{(s_h,a_h)\sim\pi} [b_h(s,a,\mathcal{D}_h)] \leq \frac{1}{N} \sum_{n=1}^N \mathbb{E}_{(s_h,a_h)\sim\pi} \left[ b_h(s,a,\mathcal{D}_h^{(n-1)}) \right]$$

$$\leq \frac{1}{N} \sum_{n=1}^N b_h(s_h^n, a_h^n, \mathcal{D}_h^{(n-1)}) + \mathcal{O}\left( H\sqrt{\frac{\log(1/\delta)}{N}} \right)$$

$$\leq \mathcal{O}\left( H\sqrt{\frac{d_E(1/N, \mathcal{F}_h)\beta}{N}} + H\sqrt{\frac{\log(1/\delta)}{N}} \right),$$

where the second inequality uses Azuma-Hoeffding inequality and the last one uses the standard regret guarantee for eluder dimension (e.g., Lemma 2 in Russo and Van Roy [2013]).

