# OpenReview forum: "Optimistic Natural Policy Gradient: a Simple Efficient Policy Optimization Framework  for Online RL"
_NeurIPS.cc/2023/Conference — NeurIPS 2023 spotlight_

### Official Review · Reviewer_czwS · 2023-07-03

**Soundness:** 4 excellent
**Presentation:** 4 excellent
**Contribution:** 3 good
**Rating:** 8
**Confidence:** 4

**Summary:**

This paper proposes a simple and efficient policy optimization method with rigorous theoretical guarantee. The algorithm combines the natural gradient ascent and the optimistic off policy evaluation and is computationally efficient. They provide the condition under which the algorithm is guaranteed to be efficient and can provably learn the optimal policy. The authors also gave three examples to show the efficiency of the algorithm: tabular case, linear function approximation and general function approximation. They show that their algorithm achieves the optimal dependence on d for linear case which improves the algorithm proposed by Zanette et al, 2021 by a factor of d. They also show that their algorithm is the first one to achieve polynomial sample complexity in general function approximation case.

In general, the proof is sound and also the writing is very good and clear. I like the simplicity of their algorithm.

**Strengths:**

1. The algorithm is simple, efficient and easy to understand.
2. They provide a sound argument of theory and the proof is sound to my knowledge.
3. In linear case, their algorithm achieves the optimal dependence w.r.t. the dimension d with computational efficiency.
4. Besides 3, they also provide the sample complexity in tabular and general function approximation case.

**Weaknesses:**

I do not think this paper has any apparent weakness. Though, I still have some minor questions about it.

1. In Theorem 1, you claim the result holds with probability at least 1/2, and in Line 207-211,  you claim that you can achieve the probability 1-\delta for any \delta with a sample complexity multiplies by a log factor. I understand that you only need to run the algorithm one for log(1/\delta) times, but how do you 'estimate the values of every output policy to accuracy \eps'? Can you show exactly how you do to achieve this? In this estimation stage, do you need to sample additional online data using the output policy or not?

2. In line 223-226, you claim the best sample complexity for policy optimization in the tabular case is H^4 S A / \eps^2. I am not sure whether this is true. Indeed, the Wu et al. you cited expressed their result in the form of regret, and you can of course uniformly sample one policy from the policies output in Wu et al and get the sample complexity. However, in this paper (https://arxiv.org/pdf/2007.13442.pdf), they propose the algorithm BPI-UCBVI for best policy identification (another name of 'policy optimization'?), and they showed the sample complexity to be H^3 S A / \eps^2 log(1/\delta). (This is the second algorithm in their paper and the first one is reward-free version. Even the traditional UCBVI algorithm should be able to achieve the H^3 S A / \eps^2 * polylog(1/\delta). I wonder whether they considered a different setting as you did and whether result you claimed is the best one.

3. In all three subroutines about OPE, you split the data into H even disjoint subset. Do you think this splitting is inefficient? I remember in some papers, such as this one (https://arxiv.org/pdf/2106.04895.pdf), they split the data into three instead of H subsets to reduce the sample complexity. Do you think this will help you further reduce the sample complexity in your algorithm?

**Questions:**

/

**Limitations:**

a typo:
Line 284: in the worse case --> in the worst case.

---

> ### Author Rebuttal · Authors · 2023-08-09
>
> We thank the reviewer for the valuable and positive feedback.
> Please see our response below.
>
> **Q1.** I understand that you only need to run the algorithm one for $\log(1/\delta)$ times, but how do you estimate the values of every output policy to accuracy $\epsilon$? Can you show exactly how you do to achieve this? In this estimation stage, do you need to sample additional online data using the output policy or not?\
> **A1.** We can evaluate  these
> $\log(1/\delta)$  policies by running each of them for $\tilde{\mathcal{O}}(H^2/\epsilon^2)$ episodes. Then  the one with the highest empirical estimate is selected.
> Therefore, the reviewer is right that additional online data is needed. Nonetheless, the amount of additional data is only $\tilde{\mathcal{O}}(H^2/\epsilon^2)$, which is smaller than the sample complexity of optimistic NPG.
> As a result, the overall sample complexity remains the same order up to logarithmic terms.
>
>
> **Q2.** The best sample complexity for policy optimization in the tabular case.\
> **A2.** Thank you for catching the typo. By regret-to-PAC conversion,  the result in Wu et al., (2022) implies a sample complexity of order $\tilde{\mathcal{O}}(H^3SA/\epsilon^2)$ that matches the lower bound for tabular MDPs up to logarithmic terms. We will fix this typo in the revision.
> Thank you for pointing out the  BPI-UCBVI algorithm. As we perceive it, the algorithmic framework of BPI-UCBVI bears more resemblance to  the UCBVI algorithm (Azar et al., 2017), as mentioned in Section 4.1 of the BPI-UCBVI paper.  Specifically, the policies in BPI-UCBVI are always chosen to be the greedy policies with respect to the Q-estimate. Therefore, in our understanding it leans more towards a value-based methodology instead of a policy optimization one.
>
> **Q3.** Can we use the reference-advantage decomposition to avoid $H$-fold data splitting?\
> **A3.** This is a great observation. Upon initial consideration, we find this idea very promising.  While extending the three-fold splitting technique proposed in Xie et al. (2022) from tabular MDPs to linear MDPs, the only part that we feel uncertain about is how to construct the Bernstein-style bonus (i.e., $b_{h,1}(s,a)$ in Line 7 of Algorithm 1 in Xie et al., 2022) for linear MDPs. We believe that the Bernstein-style bonus introduced in Pu et al., (2023) could potentially serve our purpose. However, a meticulous and thorough analysis is required in determining the viability of this approach.
>
>
>
> **Q4.** a typo: Line 284: in the worse case $\rightarrow$ in the worst case.\
> **A4.** Thank you for spotting the typo. We will fix it in the revision.
>
> ---
> Pu et al., (2023). Nearly Minimax Optimal Reinforcement Learning with Linear Function Approximation.

---

> > ### Comment · Reviewer_czwS · 2023-08-17
> >
> > Thanks the authors for answering my questions! This is a very good paper. I will keep the positive score.

---

### Official Review · Reviewer_LFfR · 2023-07-06

**Soundness:** 4 excellent
**Presentation:** 4 excellent
**Contribution:** 3 good
**Rating:** 7
**Confidence:** 3

**Summary:**

The paper presents $\texttt{OPTIMISTIC NPG}$, a new algorithm that combines the natural policy gradient with optimistic policy evaluation to encourage exploration in online RL. The algorithm demonstrates computational efficiency and achieves optimal sample complexity in linear MDPs, outperforming existing state-of-the-art approaches. Furthermore, it extends to general function approximation, making it the first algorithm to achieve polynomial sample complexity for learning near-optimal policies in this broader setting.

**Strengths:**

- The authors propose a computationally efficient policy optimization algorithm with improved sample complexity in $d$-dimensional linear MDPs.
- The authors Introduce the first on-policy method in the exploration setting that achieves polynomial sample complexity, contrasting with previous off-policy algorithms.
- The authors extend the linearMDP algorithm to a general function approximation algorithm, exhibiting polynomial sample complexity measured by the eluder dimension of a function class.

**Weaknesses:**

- The technical novelty of the algorithm for a general function class appears unclear, and further explanation is needed on how to construct confidence sets when working with function classes beyond linear functions.
- Excluding the issue of computational intractability, Subroutine 3 appears to be applicable to general function classes with bounded eluder dimension. For example, Dong et al. (2021) proved that "one-layer neural nets do not have polynomially-bounded eluder dimension," presenting an exponential eluder dimension. It is challenging to claim that the algorithm proposed in this paper is sample efficient for function classes with such characteristics.

\--

Dong, K., Yang, J., & Ma, T. (2021). Provable model-based nonlinear bandit and reinforcement learning: Shelve optimism, embrace virtual curvature. Advances in Neural Information Processing Systems, 34, 26168-26182.

**Questions:**

- On line 138, there appears to be a typo in the equation where $\psi$, a $d$-dimensional vector, is integrated with respect to s, resulting in a real number bound.
- In comparison to traditional off-policy policy gradient algorithms, what practical advantages does the on-policy algorithm offer?

**Limitations:**

Since this is a theoretical paper, its direct societal impact seems to be neutral.

---

> ### Author Rebuttal · Authors · 2023-08-09
>
> We thank the reviewer for the valuable and positive feedback.
>
>
> **Q1.** The technical novelty of the algorithm for a general function class appears unclear, and further explanation is needed on how to construct confidence sets when working with function classes beyond linear functions.\
> **A1.**
> The mathematical definition of the confidence set can be found in Line 6-7 of Subroutine 3 (General OPE) while additional intuitive explanations are offered in Line 257-268.  We kindly request the reviewer to specify the aspects of the confidence set construction that appear unclear, and we are enthusiastic about the opportunity to provide further clarification and address any points of confusion.
>
>
>
> **Q2.** Dong et al. (2021) proved that ''one-layer neural nets do not have polynomially-bounded eluder dimension," presenting an exponential eluder dimension. It is challenging to claim that the algorithm proposed in this paper is sample efficient for function classes with such characteristics.\
> **A2.** Dong et al. (2021) also established that ''the minimax sample complexity of one-layer neural nets with ReLU activation is $\Omega(\epsilon^{-(d-2)})$'' (refer to Theorem 5.1 in  Dong et al. (2021)).
> Consequently, the fact that ``one-layer neural nets do not have polynomially-bounded eluder dimension" should not be viewed as a  drawback of  this work, because information-theoretically any RL algorithm requires at least  $\Omega(\epsilon^{-(d-2)})$ samples to effectively learn one-layer neural nets under worst-case scenarios.
>
> **Q3.** On line 138, there appears to be a typo in the equation where   $\psi$, a  $d$-dimensional vector, is integrated with respect to $s$, resulting in a real number bound.\
> **A3.** Thank you for spotting the typo. The correct version should be the $\ell_2$-norm of the left hand side is bounded by the same scalar. We will fix it in the revision.
>
>
> **Q4.** In comparison to traditional off-policy policy gradient algorithms, what practical advantages does the on-policy algorithm offer?\
> **A4.**
> In the realm of empirical RL, many widely-used policy gradient algorithms (e.g., PPO and TRPO) are on-policy.
>  On-policy algorithms typically demand less memory resources because they do not  maintain an off-policy dataset as in off-policy algorithms (e.g., the replay buffer in A2C).
> Regarding performance, the superiority between on-policy and off-policy methods hinges on the specific tasks at hand. For instance,  Figure 6 in Schulman et al. (2017) highlights a comparison among PPO (on-policy), A2C (off-policy), and ACER (off-policy) across all 49 ATARI games featured in OpenAI Gym. The results therein illustrate that these algorithms excel in distinct tasks, thereby underlining the task-dependent nature of their effectiveness.
>
>
> ---
> Schulman et al., (2017). Proximal Policy Optimization Algorithms.

---

> > ### Comment · Reviewer_LFfR · 2023-08-14
> >
> > Thank you for the detailed response. It is helpful in understanding the paper. As a result, I have adjusted the scores accordingly.

---

### Official Review · Reviewer_1VAU · 2023-07-07

**Soundness:** 4 excellent
**Presentation:** 4 excellent
**Contribution:** 3 good
**Rating:** 7
**Confidence:** 4

**Summary:**

This work proposes a simple-efficient efficient policy optimization algorithm for online RL. When specialized to linear MDPs, the algorithm improves the best known result by a factor of $d$. Moreover, this is the first efficient policy optimization algorithm under general function approximation.

**Strengths:**

1 This work is well-written and well-organized.

2 This work showcase a generic policy optimization meta-algorithm can be sample efficient.

3 The theoretical guarantee outperforms the exisiting result in linear MDPs, and is the first sample efficient policy optimiztion algorithm under general function approximation.

**Weaknesses:**

1 When specialized to tabular scenario, the result is worse than the best existing result.

2 Under linear MDPs, the dependency on $\epsilon$ is not optimal.

**Questions:**

1 When specialized to linear MDPs, the algorithm is computationally efficient. When specialized to general function approximation, the algorithm is not computationally efficient due to the hardness of selecting optimitic Q-function in the confidence set. From what I know, [Ruosong Wang et al.] claims they developed a provably efficient (both computationally and statiscally) Q-learning algorithm. My question is that for the policy optimization under general function approximation, is it possible to devise a computationally efficient algorithm under the proposed optimistic NPG framework?

2 Can the authors comment on the subtle technical challenge in [Cai et al.] (linear mixture MDPs) in dealing with linear MDPs? Is the difficulty caused by covering number argument?

[Ruosong Wang et al.] Reinforcement Learning with General Value Function Approximation: Provably Efficient Approach via Bounded Eluder Dimension

---

> ### Author Rebuttal · Authors · 2023-08-09
>
> We thank the reviewer for the valuable and positive feedback.
> Please see our response below.
>
>
> **Q1.** From what I know, [Ruosong Wang et al.] claims they developed a provably efficient (both computationally and statiscally) Q-learning algorithm. For policy optimization under general function approximation, is it possible to devise a computationally efficient algorithm under the proposed optimistic NPG framework?\
> **A1.**  The computational efficiency in Wang et al., (2020) is established based on the following assumption: given a state-action pair $(s,a)$, a set of state-action pairs $\\{ (s^i, a^i) \\}\_{i=1}^n$ and $\epsilon>0$, one can  determine whether $(s,a)$ is $\epsilon$-dependent on  $\\{ (s^i,a^i) \\}\_{i=1}^n$ with respect to function class $\mathcal{F}$ in a computationally efficient way.
> As it stands, it is unclear when this assumption holds except for
> the case $\mathcal{F}$ being (generalized) linear function class.
> Consequently, the question of designing computationally efficient algorithms for general function approximation remains an open challenge, especially when considering reasonable computation-oracle assumptions.
> However, it's noteworthy that one can obtain computationally efficient policy optimization (PO) algorithms by amalgamating our optimistic NPG framework with the bonus-design and computation-oracle assumption used in Wang et al., (2020).
>
>
> **Q2.** Can the authors comment on the subtle technical challenge in Cai et al., (2020)  (linear mixture MDPs) in dealing with linear MDPs? Is the difficulty caused by covering number argument?\
> **A2.** Yes, the difficulty is caused by controlling the covering number of all the possible V-estimates. In each $k$-th iteration, Cai et al., (2020) need to control an estimation error of form $\left| \mathbb{E}\_{s'\sim\mathbb{P}\_h(s,a)} [r\_h(s,a)+V\_{h+1}^k(s')] - Q\_h^k(s,a) \right|$ where $Q\_h^k(s,a)$ is estimated by using  dataset $\mathcal{D}\_h:=\\{(s\_h^i,a\_h^i,r\_h^i,s\_{h+1}^i)\\}\_{i=1}^n$ and $V_{h+1}^k$ (the $V$-estimate at step $h+1$), e.g., ridge regression. In Cai et al., (2020),  $\mathcal{D}\_h$ consists of all the historically collected data, including those that are correlated with $V\_{h+1}^k(\cdot)$. This creates undesired dependence between $\mathcal{D}\_h$ and $V\_{h+1}^k(\cdot)$, which hinders standard concentration.  One natural way to fix this problem is perform uniform concentration for all possible $V\_{h+1}^k(\cdot)=\mathbb{E}\_{a\sim \pi^k\_{h+1}(\cdot)}[Q\_{h+1}^k(s,a)]$, which requires bounding the log-covering number of all possible $V\_{h+1}^k(\cdot)$.  Unfortunately, so far it remains unclear how to get any bound better than $\tilde{\mathcal{O}}(kd^2)$ that is the log-covering number of all possible $\pi^k\_{h+1}(\cdot)$. As a result, it remains open whether the algorithm in Cai et al., (2020)  can generalize to linear MDPs or not.

---

> > ### Comment · Reviewer_1VAU · 2023-08-16
> >
> > I thank the authors for their response. I am keeping my score.

---

### Official Review · Reviewer_EF4H · 2023-07-23

**Soundness:** 3 good
**Presentation:** 4 excellent
**Contribution:** 3 good
**Rating:** 7
**Confidence:** 3

**Summary:**

This paper proposes optimistic NPG for online RL, which is computationally efficient and enjoys the polynomial sample complexity while learning near-optimal policies. The paper is well-written and organization is clear.

**Strengths:**

There are several strengths:
1. Sample complexity benefits: compared with other computationally efficient policy optimization algorithms, the proposed algorithm in this work has better sample complexity guarantees; (I think this is the major strength)
2. The first polynomially sampled policy optimization algorithm under general function approximation.
3. Literature comparison and position is clear. The paper is well-written!

**Weaknesses:**

I think this paper is clear and I do not have concern for now.

**Questions:**

I do not have question on this paper for now.

**Limitations:**

As the authors mentioned in the Conclusions Section, I am also curious how to avoid the factor of 1/epsilon compared with the optimal rate.

---

> ### Author Rebuttal · Authors · 2023-08-09
>
> We thank the reviewer for the valuable and positive feedback.

---

### Official Review · Reviewer_BU1M · 2023-08-01

**Soundness:** 3 good
**Presentation:** 3 good
**Contribution:** 3 good
**Rating:** 7
**Confidence:** 3

**Summary:**

This paper presents a model-free policy optimization algorithm Optimistic Natural Policy Gradient for online and episodic MDPs. The authors present sample complexity results which are better than existing results for linear MDPs. This makes it computationally efficient and optimal dimension dependence, first of its kind. The algorithm is also similar to popular RL algorithms like PPO which makes it even more interesting.

**Strengths:**

Technically solid paper with state of the art sample complexity results. The contributions are very well highlighted. The details of the algorithm are well presented by highlighting the key three modules in the algorithm. The paper also studies the online exploration scenario without access to a simulator.

**Weaknesses:**

Agarwal et. al [2020] did have some experimental results on some environments to verify indeed the proposed algorithm enjoys what is proved theoretically. In this paper, any experimental proof of "computationally efficient" is missing.

**Questions:**

1. Zanette et al [2020] were able to improve the sample complexity of FQI-style algorithm by paying the price in computation. What is that the proposed algorithm compensates on, if anything, to achieve optimal rates?


**Limitations:**

-

---

> ### Author Rebuttal · Authors · 2023-08-09
>
> We thank the reviewer for the valuable and positive feedback.  Please see our response below.
>
> **Q.** Zanette et al., (2020) were able to improve the sample complexity of FQI-style algorithm by paying the price in computation. What is that the proposed algorithm compensates on, if anything, to achieve optimal rates?
>
> **A.** Technically speaking, in each $k$-th iteration, both Zanette et al., (2020) and this work need to control an estimation error of form
> $\left| \mathbb{E}\_{s'\sim\mathbb{P}\_h(s,a)} [r\_h(s,a)+V\_{h+1}^k(s')] - Q\_h^k(s,a) \right|$
> where $Q\_h^k(s,a)$ is estimated by using  dataset $\mathcal{D}\_h:=\\{(s\_h^i,a\_h^i,r\_h^i,s\_{h+1}^i)\\}\_{i=1}^n$ and $V\_{h+1}^k(\cdot)$ (the $V$-estimate at step $h+1$), e.g., ridge regression.
> The key difference between these two works lies in how the dependence between $\mathcal{D}\_h$ and $V\_{h+1}^k(\cdot)$  is handled when performing the concentration argument.
>
> - In Zanette et al., (2020),  all the historically collected data are included in $\mathcal{D}\_h$, even those that are correlated with  $V\_{h+1}^k(\cdot)$.  This creates undesired dependence between $\mathcal{D}\_h$ and $V\_{h+1}^k(\cdot)$, which hinders standard concentration.
> However, due to the use of nested confidence set, global optimism and the FQI-style update rule, $V\_{h+1}^k(\cdot)$ always lies inside a max-linear function class  $\mathcal{F}\_{h+1}=\\{\max\_{a}\phi\_{h+1}(s,a)^{\top}\theta:~~\theta\in\mathbb{R}^d,~\|\theta\|\_2\le \sqrt{d}H\\}$. As a result, this dependence issue can be easily addressed by performing uniform concentration for all functions inside  $\mathcal{F}\_{h+1}$ whose log-covering number is only $\tilde{\mathcal{O}}(d)$.
>
> - Since this work focuses on policy optimization algorithms, our $V\_{h+1}^k(\cdot)$ admits a rather different form: $V\_{h+1}^k(s)=\mathbb{E}_{a\sim \pi^k\_{h+1}(s)}[Q\_{h+1}^k(s,a)]$. If we want to similarly use all the historical data as in Zanette et al., (2020), we need to control the log-covering number of all possible $V\_{h+1}^k(\cdot)$. Unfortunately, we could not get any bound better than $\tilde{\mathcal{O}}(kd^2)$ that is the log-covering number of all possible $\pi^k\_{h+1}(\cdot)$. As a result, we opted for an alternative approach: only using the data that is independent of $V\_{h+1}^k(\cdot)$. This subset of data approximately constitutes a fraction of $\mathcal{O}(\epsilon)$ of the complete dataset, which makes our result  worse by a factor of  $\epsilon^{-1}$ than Zanette et al., (2020) that leverages the entirety of the available data.

---

### Decision · Program_Chairs · 2023-09-21

**Decision:**

Accept (spotlight)

**Comment:**

All the reviewers agreed that the theoretical contribution of this paper to policy gradient is significant (in particular the improvement of the feature dimension). I recommend the paper be accepted for spotlight presentation.